- Relationship of Permafrost Cryofacies to Varying Surface and Subsurface Terrain
- Conditions in the Brooks Range and foothills of Northern Alaska, USA
- Andrew W. Balser, ERDC-CRREL, Fort Wainwright, AK, 99703 USA
- Jeremy B. Jones, Institute of Arctic Biology, University of Alaska Fairbanks, Fairbanks, AK,
- 99775 USA.
- 9 M. Torre Jorgenson, Alaska Ecoscience, Fairbanks, AK, 99709 USA.
- Corresponding Author: A. W. Balser, ERDC-CRREL, Fort Wainwright, AK, 99703 USA
- (Andrew.W.Balser@usace.army.mil)
- Keywords: cryostructures, terrain properties, cryostratigraphy, Brooks Range, Alaska, regional
- scale

## 15 Abstract

| 16 | Permafrost landscape responses to climate change and disturbance impact local ecology                  |
|----|--------------------------------------------------------------------------------------------------------|
| 17 | and global greenhouse gas concentrations, but the nature and magnitude of response is linked           |
| 18 | with vegetation, terrain and permafrost properties that vary markedly across landscapes. As a          |
| 19 | subsurface property, permafrost conditions are difficult to characterize across landscapes, and        |
| 20 | modelled estimates rely upon relationships among permafrost characteristics and surface                |
| 21 | properties. While a general relationship among landscape and permafrost properties has been            |
| 22 | recognized throughout the Arctic, the nature of these relationships is poorly documented in many       |
| 23 | regions, limiting modelling capability. We examined relationships among terrain, vegetation and        |
| 24 | permafrost within the Brooks Range and foothills of northern Alaska using field data from              |
| 25 | diverse sites and multiple factor analysis ordination. Terrain, vegetation and permafrost              |
| 26 | conditions were correlated throughout the region, with field sites falling into four statistically-    |
| 27 | separable groups based on ordination results. Our results identify index variables for honing          |
| 28 | field sampling and statistical analysis, illustrate the nature of relationships in the region, support |
| 29 | future modelling of permafrost properties, and suggest a state factor approach for organizing data     |
| 30 | and ideas relevant for modelling of permafrost properties at a regional scale.                         |

31

# 32 **1. Introduction**

| 33 | Permafrost landscapes are critical components of global climate change, but responses and            |
|----|------------------------------------------------------------------------------------------------------|
| 34 | feedbacks depend on ecosystem properties, which vary markedly throughout the Arctic.                 |
| 35 | Permafrost landscape structure develops through a complex interplay among climate, substrate,        |
| 36 | and surficial processes operating at multiple spatial and temporal scales (Shur and Jorgenson        |
| 37 | 2007). At the interface between the atmosphere and deep permafrost, processes of vegetation,         |
| 38 | soil, and upper-permafrost cryostructures respond to climate shifts and disturbance (Viereck         |
| 39 | 1973, ACIA 2005, Jorgenson et al. 2010a, Jorgenson et al. 2013), and mediate the influence of        |
| 40 | climate on deeper permafrost (Shur and Jorgenson 2007, French and Shur 2010). Vegetation and         |
| 41 | upper permafrost horizon development have been linked with terrain properties and climate            |
| 42 | (Kreig and Reger 1982, Shur 1988, Shur and Jorgenson 2007, Pastick et al. 2014), and are             |
| 43 | mutually influential at local and circumarctic scales, though the nature and extent of relationships |
| 44 | among vegetation and permafrost is only partially understood (Raynolds and Walker 2008,              |
| 45 | Walker et al. 2008, French and Shur 2010, Lantz et al. 2010, Kokelj and Jorgenson 2013).             |
| 46 | In the Brooks Range and foothills of northern Alaska, multiple modes of permafrost                   |
| 47 | degradation appear to be accelerating (Jorgenson et al. 2006, Bowden et al. 2008, Jorgenson et       |
| 48 | al. 2008a, Balser et al. 2009, Gooseff et al. 2009), but relationships among terrain properties,     |
| 49 | vegetation, and upper permafrost characteristics are weakly documented (Jorgenson et al. 2008a,      |
| 50 | Jorgenson et al. 2010b). Region-wide estimates of future landscape resilience and response to        |
| 51 | climate perturbation depend on spatially-explicit representations of permafrost conditions           |
| 52 | (Callaghan et al. 2004), but subsurface permafrost properties across the landscape are difficult to  |
| 53 | observe directly. Determination of permafrost properties in remote, northern Alaska depends on       |
| 54 | understanding relationships among terrain, vegetation and permafrost, and applying them at a         |

- regional scale. Determining which specific terrain properties and groups of terrain properties are
- most correlated with vegetation and upper permafrost conditions within this region, and the
- degree to which correlations apply across diverse landscapes, is central to future estimates of
- resilience, responses, and feedbacks to climate in the Brooks Range and foothills of northern
- Alaska.

#### 60 **1.1 Responses and feedbacks to climate**

Permafrost degradation rate has been increasing in recent decades throughout the

circumarctic and is anticipated to continue or accelerate (ACIA 2005, Hinzman et al. 2005,

Schuur and Abbott 2011). Marked impacts and feedbacks are expected across the cryosphere,

with shifts in ecosystem structure and function (Callaghan et al. 2004, Osterkamp et al. 2009,

Goetz et al. 2011, Myers-Smith et al. 2011), local and global hydrologic cycles (Peterson et al.

2002, Hinzman et al. 2006, Frey et al. 2007), and biogeochemistry and carbon release (Tarnocai

et al. 2009, Grosse et al. 2011, Schaefer et al. 2011).

Distinct modes of permafrost degradation correlate with specific combinations of surficial 68 landscape properties, each with a different influence on ecological, hydrological, and 69 70 biogeochemical shifts, and characterized by distinct morphologies and processes (Hinzman et al. 2005, Jorgenson and Osterkamp 2005, Schuur et al. 2009, Lafreniere and Lamoureux 2013). 71 72 Modes of permafrost degradation include active-layer deepening, as well as an array of 73 subsidence features broadly termed 'thermokarst' (Hinzman et al. 2005, Jorgenson et al. 2008a). 74 Each mode affects ecosystem properties and processes at different depths, rates, and scales, in 75 turn driving the nature and magnitude of overall impacts (Jorgenson et al. 2013). Modes of permafrost degradation in response to climate perturbation or disturbance are coupled with local 76 77 surficial conditions, including thermal properties, thaw stability, slope, hydrology and ground ice

- racteristics (Leibman et al. 2003, Lewkowicz and Harris 2005, Jorgenson et al. 2008a, Kokelj
- ret al. 2009, Jorgenson et al. 2010a, Lantuit et al. 2012). Thermal properties, thaw stability, and
- hydrology, in turn, are influenced by cryostructure distribution and ground ice content,
- vegetation, and soil composition and organic layer development (Shur and Jorgenson 2007).

#### 82 **1.2 Landscape variability**

Vegetation development on the surface and cryostructure development in the upperpermafrost are dynamically linked ecosystem processes organized in complex but potentially 84 generalizable patterns across landscapes. Mutual influences between vegetation and permafrost 85 86 (Raynolds and Walker 2008) are linked with terrain characteristics, surficial thermal properties, and hydrology (Shur and Jorgenson 2007, French and Shur 2010). These may be considered 87 within the 'state factor' framework, which groups terrain properties within five umbrella 88 89 categories: biota, parent material, topography, climate, and time (Jenny 1941, van Cleve et al. 90 1991, Jorgenson et al. 2013).

On newly deposited surfaces, topography, surficial geology, climate, and potential 92 recruitment drive initial development of vegetation and new cryostructures, and influence the 93 fate of pre-existing ground ice, such as relict glacial ice (Washburn 1980, Shur 1988, Walker et al. 2008, French and Shur 2010). With time, vegetation and cryostructure development exert 94 95 increasing influence at the surface, mediating heat flux, soil moisture, and decomposition rate of organic matter, which in turn feeds back on vegetation and cryostructure development (Davis 96 97 2001, Hobbie and Gough 2004, Walker et al. 2008). Vegetation, active-layer depth, and the nature and degree of permafrost and cryostructure development across heterogeneous landscapes 98 are a product of these interactions (Shur and Jorgenson 2007, Raynolds and Walker 2008, 99 100 Walker et al. 2008, French and Shur 2010, Walker et al. 2011). Correlations among vegetation

- and permafrost characteristics are recognized from studies at specific sites (Kreig and Reger
- 1982, Shur and Jorgenson 2007, Walker et al. 2008, Kanevskiy et al. 2011, Epstein et al. 2012),
- and from regional to circumarctic-scale studies (Raynolds and Walker 2008, Gruber 2012,
- Pastick et al. 2014).
- **1.3 Integrating terrain properties**

A general approach describing relationships among terrain properties and permafrost,

congruent with the state factor framework (Shur and Jorgenson 2007), has been developed to

better estimate permafrost vulnerability among different landscapes. Terrain properties and

permafrost characteristics co-vary, and consistency of associations among permafrost, terrain and

vegetation enable landscape-scale analysis on that basis (Jorgenson and Kreig 1988, Raynolds

and Walker 2008, Jorgenson et al. 2010a, Jorgenson et al. 2013, Pastick et al. 2014). While the

112 importance of surficial deposits (Kreig and Reger 1982, Jorgenson et al. 2008a) and vegetation

(Viereck 1973) to ground ice and permafrost development have long been recognized,

landscape-scale methods for integrating terrain factors are not fully developed. Toward improved

terrain factor integration, we hypothesized that: 1) vegetation and permafrost properties

consistently correlate with specific terrain conditions across landscapes due to these

relationships; 2) that diverse landscapes may fall into general groupings from statistical analysis

of empirical field data for these combined properties; and 3) that these relationships can be used

to help identify which terrain factors, in combination, facilitate spatial characterization of

surficial landscape properties in the Brooks Range and foothills of northern Alaska.

Our research tested these ideas statistically using ordination of field survey data collected

from sites representing diverse landscapes in the Brooks Range and foothills of northern Alaska.

Identifying statistically-supported linkages between permafrost properties (ground ice content

- and cryostructures), and terrain properties (vegetation and surficial geology), can facilitate
- regional scale estimation of permafrost vulnerability and estimation of ground ice conditions,
- and better inform models examining regional resilience, response and feedbacks to climate
- change.
- **2. Methods**
- 2.1. Study region

Our research spanned a gradient of arctic tundra including barren, herbaceous, and shrub landscapes within Alaska's Brooks Range and foothills, from the east-central portion of Alaska's 132 133 North Slope westward through the Noatak Basin to the Mission Lowlands, near the Noatak delta (Figure 1). These periglacial landscapes are within the continuous permafrost zone (Jorgenson et 134 al. 2008b) and are part of Arctic Bioclimate Subzone E (CAVM-Team 2003). The northeast 135 136 portion of the study region was centered around Toolik Field Station on the north slope of Alaska, while the central and western study region followed the Noatak Basin from near its 137 headwaters downstream to the Mission Lowlands, near the Noatak River delta. 138 139 Toolik Field Station is located in the northern Brooks Range foothills within a mosaic of

landscapes of varying glacial ages and ecotypes. Physiography ranges from low mountains at 141 the edge of the Brooks Range to subtle foothills stretching more than 75 km from the mountains 142 to the edge of the Arctic Coastal Plain. Date since most recent glaciation ranges from early 143 Pleistocene to Holocene for field sites surrounding Toolik Field Station, with acidic and 144 nonacidic, graminoid and shrub tundra vegetation reflecting duration of ecosystem development 145 and local site conditions (Walker et al. 1994, Walker et al. 1995, Hamilton 2003, Walker and

| 146 | Maier 2008). Lake and stream density is variable by landscape age-class and related with glacial           |
|-----|------------------------------------------------------------------------------------------------------------|
| 147 | and periglacial landforms (Hobbie et al. 1991, Kling 1995, Hamilton 2003).                                 |
| 148 | The Noatak River flows 730 km along a westward course at approximately $67.5^{\circ}$ N (Figure            |
| 149 | 1). Most of the 33,100-km <sup>2</sup> basin falls within the Noatak National Preserve (U.S. National Park |
| 150 | Service) and is recognized as a UNESCO Biosphere Reserve. The Noatak Basin was                             |
| 151 | periodically glaciated throughout the Pleistocene and contains a patchwork of glacial and                  |
| 152 | periglacial landforms ranging in age from early Pleistocene to contemporary (Hamilton 2010,                |
| 153 | Hamilton and Labay 2011). Physiographic provinces include high mountains of the east-central               |
| 154 | Brooks Range, through foothills and valley bottoms to the Mission Lowlands at the arctic-boreal            |
| 155 | ecotone near the Noatak mouth (Wahrhaftig 1965, Young 1974). Land cover spans a gradient of                |
| 156 | vegetation and ecotypes including arctic and alpine tundra, shrublands and lowland boreal forest           |
| 157 | (Young 1974, Viereck et al. 1992, Parker 2006, Jorgenson et al. 2010b).                                    |
| 158 | Landscape conditions throughout this 500-km-wide region represent a broad range of                         |
| 159 | typical low-arctic landscapes (Figure 2). Alpine, foothill, and valley bottom settings include             |
| 160 | many characteristic ecotypes of the North American Low Arctic, a suite of periglacial landforms,           |
| 161 | diverse surficial geology and lithology, and a broad continuum of permafrost characteristics and           |
| 162 | cryostructures. While a geographic gap exists between the Toolik and Noatak subregions,                    |
| 163 | substantial overlap among terrain properties and permafrost cryostratigraphy link them                     |
| 164 | conceptually. Our study deliberately included a wide range of conditions over a large                      |
| 165 | geographic area to represent a diversity of low-arctic landscapes in the region.                           |
| 166 | 2.2 Field surveys                                                                                          |

| 167 | Our regional surveys identified areas of surface-exposed and degrading permafrost                  |
|-----|----------------------------------------------------------------------------------------------------|
| 168 | distributed among diverse landscapes, from which we selected field sites representing a range of   |
| 169 | low-arctic conditions. Aircraft-supported field campaigns and airphoto analysis in 2006, 2007,     |
| 170 | and 2008 were used to identify watersheds with actively degrading permafrost exposures             |
| 171 | representing different modes of degradation (and by proxy, differing ground-ice conditions).       |
| 172 | Several thousand permafrost degradation feature locations were recorded in an ArcGIS               |
| 173 | GeoDatabase, which was later expanded and augmented through a subsequent National Park             |
| 174 | Service survey, which included both Gates of the Arctic National Park and Preserve and Noatak      |
| 175 | National Preserve, using high-resolution satellite imagery to census these features throughout     |
| 176 | both park units (Balser et al. 2009, Swanson and Hill 2010). These data drove spatial analyses     |
| 177 | identifying diverse combinations of ecotype, lithology and surficial geology among                 |
| 178 | subwatersheds accessible by helicopter from field camps at Kelly River, Feniak Lake, and Toolik    |
| 179 | Field Station (Figure 1). During subsequent helicopter-based visits in 2009, 2010, and 2011,       |
| 180 | field sites were chosen for detailed examination based on: 1) best accessibility to exposures of   |
| 181 | permafrost; and 2) inclusive representation among terrain properties including ecotype, lithology, |
| 182 | and surficial geology.                                                                             |
|     |                                                                                                    |

At each of 54 field sites, we measured and described general landscape characteristics and specific conditions at the site of permafrost exposure. A subset of categorical and quantitative data collection protocols and field codes were adopted from Jorgenson et al. (2010b) to characterize ambient surface properties (within approximately 100 m of the permafrost exposure) and to catalog the specific combination of vegetation, soil, surficial geology and cryostratigraphy immediately at the site of permafrost exposure (Table 1). Basic geomorphology, lithology, surficial geology, topography, and landforms were recorded to represent the area within

- approximately 100 m of the permafrost exposure. Vegetation was recorded both by class
- (Viereck et al. 1992) and as a list of predominant overstory and understory species of vascular
- plants, and functional groups of bryophytes within 20 m of the permafrost exposure.
- Permafrost profile exposures were described in detail to characterize and quantify 194 properties of the live vegetative mat, contemporary soil (organic and mineral), parent material 195 and archaic soils, coarse fraction, ice content, cryostratigraphy, and interpretations of 196 mechanisms of cryogenesis. Permafrost exposures were predominantly composed of vertical scarps at actively degrading edges of retrogressive thaw slumps, active layer detachment slides 197 198 and thermo-erosional gullies (Figure 3). Permafrost exposures were prepared using hand tools to 199 remove previously thawed material and expose an intact permafrost profile from the top (ground 200 surface) down to the greatest accessible depth within the thaw feature (Figure 4). Exposures were prepared to a width of at least 1 m, with categorical and quantitative tabular data taken for 201 202 each discernible layer in the profile (Figure 4) from vegetation at the surface to the bottom of the 203 exposure. Data from each discernible subsurface layer were weighted by layer thickness and 204 integrated to generate overall values for: 1) contemporary soil; and 2) archaic soil layers and parent material (Table 2). Hand-drawn cryostratigraphic maps roughly following Kanevskiy et 205 al. (2011), and detail photos for each permafrost profile complement data and general site photos 206 207 and were used for interpretation and summarization.
- 2.3 Data analysis
- **2.3.1 Data reduction**

To statistically analyze data from our 46 field sites by terrain and permafrost properties, we began with data reduction to eliminate extraneous independent variables with minimal contribution to our model and to reduce redundancy in the data. We employed Pearson (r)

| 213 | correlations in two separate steps to examine redundancy and to identify variables with minimal        |
|-----|--------------------------------------------------------------------------------------------------------|
| 214 | contribution to ordination results. In the first step, a Pearson correlation analysis of all variables |
| 215 | against one another with R statistical software was used to examine inter-variable relationships       |
| 216 | and identify groups of variables that might be represented by a single integrator variable. Where      |
| 217 | a set of variables was grouped by Pearson scores $> 0.60$ for all pairings, the group was              |
| 218 | considered a candidate for integration.                                                                |
| 219 | In the second step, all variables went through a pilot, three-axis non-metric                          |
| 220 | multidimensional scaling (NMS) ordination with 50 runs of 250 iterations in PC-ORD to                  |
| 221 | generate Pearson correlation values for each variable against each ordination axis. This               |
| 222 | ordination was used to examine the contribution of each variable to the ordination and eliminate       |
| 223 | those with minimal analytical value. For this analysis, categorical data were transformed to           |
| 224 | binary numbers for each categorical unit of each categorical variable, while continuous and            |
| 225 | ordinal data were scaled 0 to 1 (precision to the hundredth) to conform with NMS analysis              |
| 226 | assumptions for a valid distance matrix (McCune and Grace 2002, McCune 2013). Those                    |
| 227 | variables with NMS Pearson scores $

- traditional ordination (e.g., site similarity and clustering in multidimensional space as determined
- by a distance matrix), our dataset comprised different logical groupings of data for each site
- 238 (e.g., ice, substrate and vegetation) and dissimilar data types, such as coarse fragment size class
- 239 (ordinal), vegetation type (categorical), and ice percentage (continuous).

240 MFA, a recent adaptation of principal component analysis (PCA), was chosen for this 241 application of ordination because it is designed to integrate dissimilar data types and different logical groupings of data (termed 'blocks') for each observation within a single ordination run 242 (Escofier and Pagès 1994). While other ordination techniques, such as NMS, can also be applied 243 244 after data transformation and scaling (McCune and Grace 2002, McCune 2013), MFA offers two distinct advantages over NMS and other ordination techniques under these conditions. First, 245 end-user data transformation is unnecessary because MFA performs data normalization in an 246 247 initial PCA step, using the square root of the first eigenvalue in a manner comparable to Z-score 248 normalization (Abdi et al. 2013). These normalized data are then merged to form the analysis 249 matrix, enabling valid distance matrices to be calculated from what were initially incongruous 250 variables. Second, MFA provides the option to define blocks of data, which are conceptually coherent groups of variables pertaining to all observations (Abdi et al. 2013). The chief 251 advantage of a block approach is that individual blocks of data (e.g., vegetation, substrate, ice) 252 253 are inhibited from dominating the ordination results while other blocks become de-emphasized. MFA achieves this parity by normalizing the input data by block, and by handling each block as 254 255 a sub-matrix of the whole. The first principal component of each block is scaled to 1 in the 256 normalization step, which ensures that no block will dominate the model through disproportionate inertia in the final ordination (Abdi et al. 2013). Finally, each block must 257

contain variables of the same data type for the normalization step to produce valid results. Thus,

conceptual blocks containin