# Peer review of "Relationship of Permafrost Cryofacies to Varying Surface and Subsurface Terrain"

_The Cryosphere, 2016_

## Referee Comment (RC1) · Anonymous Referee #1 · 11 Dec 2016

This article investigates the relationship of subsurface soil and permafrost properties, i.e. cryostructures / cryofacies, to surface terrain properties, vegetation and modes of permafrost degradation. Four separate statistical groups are identified based on multivariate ordination and automatic classification, i.e. hierarchical clustering. The article benefits from of a very interesting and promising dataset on permafrost characteristics in association with surface properties and degradation modes. This topic is of general interest and addresses a current research need. The analysis of the dataset using multivariate ordination statistics hints some interesting results. Overall the article shows some promising insights well suitable for publication in The Cryosphere. Yet, there are

several major issues that need to be addressed.

My first concern is the multivariate statistical analysis that should be reevaluated. I don't understand why the authors perform a data-reduction before using non-metric multidimensional scaling (NMS) and MFA. NMS should cover these redundancies and you describe that PC-ORD is running person correlation iteratively. Also, principal component analysis (PCA) on its own is a data reduction method. I am concerned that a lot of information is lost along the way. If you perform these steps, you should cite literature to support your workflow. Also, the MFA results show to some extent a so called horseshoe artifact. It seems to me that this analysis is overly complicated and the article lacks a second strong line of analysis to make sense of the collected data.

The discussion on state factors is not well developed in the analysis. Only two state factors are addressed (biota and parent material). Consider removing state factors or expand significantly on this to draw conclusions on this topic. What about climate, topography and time? The data is there. For instance, topography and climate are partly available from Table 3.

The language of the article is of very good quality, but the article suffers from several repetitions in the text that lead to unnecessary length in relation to the content. Existing sections could be shortened by 20%. The authors could be more concrete in their discussion and how their findings compare to studies by others. Consider this for the relationship of surface and subsurface properties, grouping/classification of permafrost terrain and potentially a discussion beyond the regional extent.

The theme of permafrost degradation is touched several times, but only loosely connected and integrated throughout the text. Table 5 is not referenced in the text. This could be a strong point of the article and I encourage to develop this further and I am looking forward to see the results.

Minor comments: L 77 Ground ice as a surficial property? L 76-81 This needs to be condensed and presented clearer. L 118 rephrase the third hypothesis L 134 I think

it is more appropriate to cite Brown 1997 instead of Jorgenson et al. 2008b. L 170 Why no areas without active permafrost degradation? This way you cannot distinguish between stable and unstable areas and the factors governing them. Something to consider for the future. L 195 Archaic soils? I presume you mean paleosols (the term used in soil taxonomy). L 209ff You have to cite where you found this workflow and find literature that justifies its use as a preliminary step for NMS and MFA. L 226 I am not sure if you can cite personal communications as a separate reference. L 226 What distance matrix/metric have you used? Euclidean? Please motivate your choice. L 253 Shouldn't it be 'emphasized' instead of 'de-emphasized'? Rephrase. L 243 and 245 repetition of 'other ordination techniques' L 260ff this paragraph can be condensed. L 300 Please use short headings for these groups to make them better understandable. e.g. Group E1 – typically late-Pleistocene moraine deposits L415 Could the removal be avoided by data-transformation? L423 Please add a paragraph on the suitability of your methods to investigate this specific problem and potential sources of error and misinterpretation, such as the horseshoe artifact in Fig. 5. Also, what groupings/classes have other researchers found or suggested for permafrost environments and how can you relate these to your results? L 469 What correlations do you mean exactly and what correlations have others found? This should be expanded and more reference to existing literature from other study areas would be helpful to draw general conclusions. L 480-493 Please be more specific in your comparison to other studies. What correlations have others found? How does this compare to yours and what are the underlaying mechanisms? Are there any concrete examples?

Fig. 6: What do these abbreviations mean? Please spell them out. These graphs are not color-blind friendly. Please use different symbols for each of the groups.

Fig. 7: What does Inertia mean?

Fig. 9: B should always show the landscape, not only the surface.

Table 5 and Table 6 are not referenced in the text.

Does the paper address relevant scientific questions within the scope of TC? 1. Does the paper present novel concepts, ideas, tools, or data? Yes, but not yet convincing 2. Are substantial conclusions reached? They need to be better undermined 3. Are the scientific methods and assumptions valid and clearly outlined? I outlined some concerns 4. Are the results sufficient to support the interpretations and conclusions? Partly 5. Is the description of experiments and calculations sufficiently complete and precise to allow their reproduction by fellow scientists (traceability of results)? yes 6. Do the authors give proper credit to related work and clearly indicate their own new/original contribution? Partly 7. Does the title clearly reflect the contents of the paper? yes 8. Does the abstract provide a concise and complete summary? yes 9. Is the overall presentation well structured and clear? yes 10. Is the language fluent and precise? partly 11. Are mathematical formulae, symbols, abbreviations, and units correctly defined and used? - 12. Should any parts of the paper (text, formulae, figures, tables) be clarified, reduced, combined, or eliminated? I guess some tables could be combined and moved to the supplement. 13. Are the number and quality of references appropriate? References after 2014 are missing. 14. Is the amount and quality of supplementary material appropriate? -

---

## Referee Comment (RC2) · Anonymous Referee #2 · 13 Jan 2017

I have read this manuscript with great interest. I have personal knowledge of the area within the Brooks Range and foothills of northern Alaska through short visits but no research activities of my own. I think I learnt a lot from this interesting paper. The authors investigate and discuss the relationships between terrain, vegetation and permafrost using an extensive set of field data from a highly representative type of sites. The relationships between permafrost and permafrost stability/instability and terrain properties are extensively investigated and discussed using a relevant multiple factor analysis. The relationships studied and discussed cover 46 well selected illustrated and described field sites. As far as I can judge the sites included in the study are well

selected and they are representative for the area and hence give a good coverage of relationships between terrain, vegetation and permafrost and its processes and status. The discussion of this paper is a nice and valuable contribution to our knowledge about this area in a global change scenario. It is also a contribution that combine parts of field work, modelling, geocryology geomorphology, ecology and climatology etc. in a nice multidisciplinary mix. As far as I can judge the paper and its subject fit well into the CRYOSPHERE Here is my summary comment with respect to the manuscript evaluation criteria; Originality (Novelty): 1 (or 2) The area and the sites chosen together with the multidisciplinary methods are well treated and innovative and this contribution will be read with interest by the scientific community – not only geocryologists. Scientific Quality (Rigour): 1 The aims and research questions are well defined and are treated with an adequate methodology. The results are discussed in a balanced way and the aims and research questions are well met, discussed and relevant conclusions are drawn. Significance (Impact): 1 The result of the study gives new knowledge regarding ground ice type and amount and also the distribution cryostructures in relation with terrain properties and vegetation at landscape and regional scales. The results of the paper clearly contribute to a better understanding of how permafrost and permafrost terrain will response to climate changes. This is highly wanted information especially in the larger regional scale. Presentation Quality: 2 The writing is generally clear. As English is not my mother tongue I have not gone into details regarding the language. The figures and pictures submitted are of high quality 1. Does the paper address relevant scientific questions within the scope of TC? Yes 2. Does the paper present novel concepts, ideas, tools, or data? Yes 3. Are substantial conclusions reached? Yes 4. Are the scientific methods and assumptions valid and clearly outlined? Yes 5. Are the results sufficient to support the interpretations and conclusions? Yes. 6. Is the description of experiments and calculations sufficiently complete and precise to allow their reproduction by fellow scientists (traceability of results)? Yes. 7. Do the authors give proper credit to related work and clearly indicate their own new/original contribution? Yes. 8. Does the title clearly reflect the contents of the paper? Yes 9. Does the abstract provide a concise and complete summary? Yes. 10. Is the overall presentation well-structured and clear? Yes. 11. Is the language fluent and precise? As far as I can judge. 12. Are mathematical formulae, symbols, abbreviations, and units correctly defined and used? Yes. 13. Should any parts of the paper (text, formulae, figures, tables) be clarified, reduced, combined, or eliminated? No. Minor Changes according to the editor might be possible. 14. Are the number and quality of references appropriate? Yes 15. Is the amount and quality of supplementary material appropriate? Yes.